# Investigation into the Potential Role of *Propionibacterium freudenreichii* in Prevention of Colorectal Cancer and Its Effects on the Diversity of Gut Microbiota in Rats

**DOI:** 10.3390/ijms24098080

**Published:** 2023-04-29

**Authors:** Ifeoma Julieth Dikeocha, Abdelkodose Mohammed Al-Kabsi, Ahmad Faheem Ahmeda, Michael Mathai, Mohammed Abdullah Alshawsh

**Affiliations:** 1Faculty of Medicine, University of Cyberjaya, Persiaran Bestari, Cyberjaya 63000, Malaysia; 2Department of Basic Medical Sciences, College of Medicine, Ajman University, Ajman P.O. Box 346, United Arab Emirates; 3Center of Medical and Bio-Allied Health Sciences Research, Ajman University, Ajman P.O. Box 346, United Arab Emirates; 4College of Health and Biomedicine, Victoria University, Melbourne, VIC 3011, Australia; 5Department of Pharmacology, Faculty of Medicine, Universiti Malaya, Kuala Lumpur 50603, Malaysia

**Keywords:** colorectal cancer, probiotics, *Propionibacterium freudenreichii*, azoxymethane, microbiome

## Abstract

Colorectal cancer (CRC) accounts for 10% of all cancer diagnoses and cancer-related deaths worldwide. Over the past two decades, several studies have demonstrated the clinical benefits of probiotic supplementation and some studies have shown that certain probiotics can modulate immunity and strengthen gut microbiota diversity. This study aims to assess the impact of the *Propionibacterium freudenreichii* (PF) probiotic against CRC induced by azoxymethane (AOM), and to investigate its effects on gut microbiota diversity in rats, as well as to evaluate the anti-proliferative activities of PF in HCT116 CRC cells. This experiment was performed using four groups of SD rats: normal control, AOM group, PF group (1 × 10^9^ CFU/mL), and standard drug control (5-fluorouracil, 35 mg/kg). Methylene blue staining of colon tissues showed that the administration of PF significantly reduced the formation of colonic aberrant crypt foci (ACF) compared to the AOM control group. In addition, treated rats had lower levels of malondialdehyde in their colon tissue homogenates, indicating that lipid peroxidation was suppressed by PF supplementation. Furthermore, 16S rRNA gene analysis revealed that probiotic treatment enhanced the diversity of gut microbiota in rats. In vitro study showed that the viability of HCT116 cells was inhibited by the probiotic cell-free supernatant with an IC_50_ value of 13.3 ± 0.133. In conclusion, these results reveal that consuming PF as probiotic supplements modulates gut microbiota, inhibits the carcinogenic effects of AOM, and exerts anti-proliferative activity against CRC cells. Further studies are required to elucidate the role of PF on the immune response during the development and growth of CRC.

## 1. Introduction

Colorectal cancer (CRC) is the third most prevalent type of cancer and the second leading cause of cancer-related fatalities globally. By 2035, it is expected that 2.5 million additional CRC cases will be diagnosed. CRC has a greater incidence and mortality rate in developed countries; however, currently, a new trend of growing CRC incidence in developing and low-income countries has emerged [1]. Various causes of CRC have been investigated using data from different cohort studies, and the results are similar to those of animal studies. The collective conclusion is that genetic risk factors and a lifestyle that may lead to overweight and diabetes contribute to a higher incidence of CRC [2]. Excessive smoking and alcohol usage, consumption of excess red and processed meats, dysbiosis, and advanced age (>50 years) are modifiable risk factors for CRC. However, there has been an alarming rise in the diagnosis of CRC in younger persons (20–50 years old), owing to cases of cancer mutation inheritance, poor cell differentiation, and the initial tumor’s position on the left side of the colon. Other risk factors, such as genetics, cannot be altered [3].

The number of studies promoting the clinical benefits of consuming probiotics has increased over the last two decades, and probiotics have been demonstrated to enhance immunity and strengthen the gut defensive line [4]. Currently, probiotics are being touted as a possible cancer-preventative method. Several RCT studies have found that CRC patients who consistently consumed fermented dairy products (containing probiotics), such as yoghurt, kefir, cheese, and probiotic tablets, showed a better outcome [5,6,7,8]. As the effect of probiotics against CRC is being studied, *Bifidobacterium* and *Lactobacillus* were the most common probiotics used in these studies.

*Propionibacterium freudenreichii* is a neutral pH, non-motile, and non-spore-forming gram-positive rod. It is a classical dairy propionibacteria that is distinct from the opportunistic pathogen *Propionibacterium acnes*, which belongs to the cutaneous propionibacteria. Recent genome-based taxonomy has led to the reclassification of classic propionibacteria, such as *Acidipropionibacterium*, *Propionibacterium freudenreichii*, and *Pseudopropionibacterium*, into a separate taxonomic category. In contrast, cutaneous bacteria have been reclassified into the genus Cutibacterium. This distinction is based on the complete genome sequencing of these bacteria [9]. Diary (classical) Propionibacteria are capable of fermenting a variety of carbohydrates, alcohol, substrates, and organic acids, which enables them to adapt to various environmental conditions [9,10].

*Propionibacterium freudenreichii* is a probiotic bacterium that has been widely used in the production of Swiss cheese. Its ability to survive the harsh conditions of cheese-making and resist digestive stress is essential to its therapeutic effects in the intestine [11]. *P. freudenreichii* has been extensively studied and has been found to meet the criteria for a probiotic, including adherence to host cells, resistance to gastrointestinal tract stressors, anticancer potential, anti-pathogenic action, immunomodulatory capabilities, and molecular characterization using omics techniques [12,13]. The probiotic properties of *P. freudenreichii* are attributed to a variety of bioactive metabolites and short-chain fatty acids (SCFAs), such as propionate and acetate, which provide energy for colon cells, stimulate the growth of normal colon cells, and restrict the growth of colon cancer cells [14]. *P. freudenreichii* is considered “generally recognized as safe” (GRAS) by the US Food and Drug Administration (USFDA), and its safety for human consumption has been evaluated by some studies, with no adverse effects reported [15,16]. Additionally, *P. freudenreichii* has an intrinsic resistance to several antibiotics, including aminoglycosides, first- and second-generation quinolones, oxacillin, metronidazole, kanamycin, and levofloxacin. This resistance is not encoded by bacterial plasmids, and no transferable antibiotic resistance has been recorded [17,18].

Several studies [19,20,21] have investigated the cancer-preventive properties of *P. freudenreichii*. For instance, Jan et al. found that *Propionibacterium* inhibits cancer cells such as Caco2, HeLa, and HT29, presumably due to its SCFA metabolites [19]. Another study revealed that *P. freudenreichii* TL133 accelerated the inhibition of cancer cells by apoptosis in the rat colon after genotoxic exposure, which was attributed to its metabolites [22].

Previous research has shown that certain gut microbiota, such as *Fusobacterium nucleatum*, *Streptococcus gallolyticus*, and *Bacteroides fragilis,* play a significant role in CRC carcinogenesis by creating physical and biological conditions that can increase the likelihood of precancerous lesions in the colon [23]. Dysbiosis of the gut microbiota contributes to the pathogenesis of CRC, and studies have suggested a specific relationship between the development of CRC and the composition of gut bacteria. By colonizing the host tissue and preventing the colonization of harmful bacteria, probiotics can help to restore gut microbial balance and counter dysbiosis [24]. This supports the notion that manipulating the gut microbiota may be a potential strategy for preventing CRC.

In this study, we investigated the anti-proliferative effects of *P. freudenreichii* metabolites on HCT116 CRC cell lines and the preventive effects of live *P. freudenreichii* cells consumed by Sprague-Dawley rats that had been induced with neoplastic lesions through the administration of azoxymethane.

## 2. Results

### 2.1. Cell-Free Supernatant of P. freudenreichii Inhibits CRC Cell Proliferation

The MTT assay results showed that the cell-free supernatant of *P. freudenreichii* decreased the viability of HCT116 cells as the dose of the PF increased. On the other hand, 5-FU had the strongest inhibitory effect against HCT116 colorectal cancer cell proliferation compared to *P. freudenreichii* (Figure 1). The IC_50_ of 5-FU against HCT116 cells after 72 h of treatment was 0.84 ± 0.02 µg/mL, while the IC_50_ of *P. freudenreichii* was 13.3 ± 0.133% at the same time point.

### 2.2. P. freudenreichii Alleviates Oxidative Stress

Malondialdehyde (MDA) concentrations in the rat’s colon samples of the AOM-induced group increased remarkably to 5.7 ± 1.9 µM/mg protein compared to the normal control (NC) group, indicating oxidative stress. On the other hand, *P. freudenreichii* probiotic supplementation reduced oxidative stress to 2.34 ± 0.87 µM/mg protein in treated rats. MDA levels were also reduced in rats given 5-FU (Figure 2).

### 2.3. Effects of P. freudenreichii on ACF Formation

Aberrant crypt foci (ACF) were found more frequently in the distal colon than in the proximal colon. The total number of crypts was determined by counting the ACF in the colon specimens using a microscope. The rats that received AOM developed recognizable lesions known as aberrant crypt foci (ACF) in the colon, as seen by methylene blue staining (Figure 3). In the colons of the normal control group, there no were abnormal crypts were observed. In comparison to the AOM control group, rats fed with probiotics had lower ACF numbers and multiplicity. The rats given *P. freudenreichii* probiotics had significantly less ACF (*p* < 0.05) compared to the AOM group. The proportion of ACF inhibition for the *P. freudenreichii* group (PF) was 56.0%, compared to 67.6% for the rats treated with the conventional medication 5-FU. The rats supplemented with *P. freudenreichii* probiotics showed a significantly reduced total ACF count 14.9 ± 2.73 (*p* < 0.001) compared to the AOM control group (33.9 ± 3.18) (Table 1).

### 2.4. Histological Findings

In the control group (AOM), colon tissues revealed dysplastic and hyperplastic crypts. These ACFs had a larger and longer mucosal lining, evident corrosion of the cell, greater inflammation, crowding of the nucleus, loss of polarity, and loss of goblet cells. The presence of ACF propagated mucosal glands in the colonic tissue portions of AOM-induced rats, as evidenced by elongated stratified nuclei, architectural atypia, frequent mitoses and mucin degradation, atypical epithelial cells, and larger-than-normal crypts when compared to normal rats. These abnormalities were alleviated to some extent in the treated groups when compared to the AOM group (Figure 4).

### 2.5. Effect of P. freudenreichii on Gut Microbiota Diversity

Firmicutes, Bacteroidota, Actinobacteriota, Proteobacteria, Verrucomicrobiota, and Patescibacteria were the most abundant phyla in each group at the phylum level, as shown by 16S rRNA sequencing. Overall, the phyla Firmicutes, Actinobacteriota, Proteobacteria, and Bacteroidetes were predominant in all groups (Figure 5).

At the genus level, the most diverse gut microbiota were found in the normal control group, followed by the *P. freudenreichii*-treated group, while the 5-FU-treated group and the AOM-induced group were the less diverse groups (Figure 6). The relative abundances of the identified taxa were visualized using a heatmap (Figure 7). Alpha diversity analysis was performed using the Shannon index to measure the diversity within the samples. A lower Shannon index indicated higher alpha diversity, whereas higher values indicated less diversity (Figure 8a). Beta diversity analysis was performed using non-metric multidimensional scaling (NMDS) to compare similarities among groups. Figure 8b,c shows that *P. freudenreichii* group diversity differs from that of the other groups. The microbiota of the control samples were less diverse than those of the other groups.

## 3. Discussion

Colorectal cancer is a prominent cause of death globally, and it is thought to be caused by the transformation of normal colonic epithelial cells into adenomatous polyps, which then turn into an invasive malignancy. Although genetics (hereditary) is the main risk factor for CRC, environmental factors such as diet and lifestyle have been linked to an increase in CRC incidence in epidemiological studies [25,26].

*Propionibacterium freudenreichii* is a probiotic-potential bacterium that is commercially useful. It is generally regarded as safe and capable of surviving digestive stress encountered by oral ingestion, making it a very interesting probiotic microorganism [10]. Through the formation of SCFAs, such as propionate and acetate, this bacterium can protect against CRC by kickstarting apoptosis [11,27].

The mitochondrial function of cells is represented by the conversion of tetrazolium salt MTT into formazan crystals that can be solubilized for homogeneous methane analysis. The decline in the cell population represents cell growth suppression in dividing cells. In our study, the cytotoxicity of *Propionibacterium freudenreichii* against colon cancer cells was assessed using the HCT116 colon cancer cell line, and our results showed that the cell-free supernatant of *P. freudenreichii* inhibits the cell proliferation of HCT116 colon cancer cells. This anti-cancer effect could be due to the release of its SCFA metabolites, such as propionic acid into the supernatant. Compared to the standard control (5FU), the inhibitory activity of 5FU against CRC cells was significantly more potent than *P. freudenreichii* (Figure 1).

In one study, *P. freudenreichii* was found to initiate apoptosis of HGT-1 human gastric cancer cells by acting on the mitochondria of the colorectal cancer cells through the metabolite propionate [28]. Furthermore, several studies have demonstrated that *P. freudenreichii* exhibits inhibitory effects on various colon carcinoma cell lines, including HT29, HeLa, and CaCo-2 cells [19,29]. Notably, *P. freudenreichii* does not display cytotoxicity or inhibit the proliferation of normal colon cells, such as Human Epithelial Intestinal Cells (HIEC) [29]. In fact, some studies have suggested that *P. freudenreichii* may have a protective effect on the intestinal epithelium by promoting cell differentiation and preventing inflammation, which can be beneficial for maintaining a healthy gut. Moreover, *P. freudenreichii* can reduce inflammation in HIEC cells, highlighting its potential as a functional food for targeting gut inflammation [30].

Aberrant crypt foci are commonly recognized as early indicators of human large intestine neoplasms. In animal experiments, ACF were induced in rats with an AOM carcinogenetic substance [31,32]. In general, CRC develops and spreads through stages, each of which features unregulated and fast cell proliferation, as well as inhibition of apoptosis [33]. The rat model captures and expresses the majority of essential molecular, morphological, and clinical characteristics associated with CRC in humans, such as the creation of ACF, which is thought to be a precursor to malignant tumors [34]. ACF development is now commonly acknowledged as a viable endpoint in short-term chemo-preventive trials, and is considered to be the gold standard of colon carcinogenesis biomarkers [33,35]. It should be noted that, although some previous studies have employed 15 mg/kg of AOM for two–four weeks [31,36], 7 mg/kg of AOM delivered to rats for three weeks was able to induce ACF development in our experiment.

In this investigation, *P. freudenreichii* was found to be beneficial in inhibiting ACF formation and reducing the frequency of preneoplastic lesions and multi-crypts in the colon of rat, which is comparable to the findings of other published studies in which probiotics suppressed chemically-induced ACF in SD rats [19,37,38]. The total count of ACF in rats treated with 1 × 10^9^ CFU/mL of *P. freudenreichii* was significantly lower, with a higher proportion of inhibition compared to the AOM group; also, in the rats treated with 5FU, there was significantly less ACF than in the AOM control group from single to multiple crypts, as seen in Table 1 and Figure 3. After staining with methylene blue, noticeable ACFs were seen in the AOM, PF, and 5FU groups (Figure 4).

The presence of atypical epithelial cells, overcrowding and elongation of the nuclei, and architectural atypia in the ACF proved that they were dysplastic aberrant crypts, which are precursor lesions of CRC. Hyperplastic aberrant crypts were also observed, which represent the first step in ACF formation before the development of mild or severe dysplasia (Figure 5).

AOM is metabolized by cytochrome P450 into a toxic metabolite (methylazoxymethanol) which causes oxidative stress, DNA mutations, and neoplastic lesions. Oxidative stress in colon cells is mediated by glutathione depletion, which impairs the total antioxidant capacity. The high rate of lipid peroxidation indicated by elevated MDA in the AOM-induced rats was reduced in the probiotic-treated rats, as probiotic supplementation reduced the reactive oxygen species. In AOM-induced rats that received no treatment, antioxidant enzymes were depleted and were unable to alleviate free radicals. The AOM group showed higher levels of lipid peroxidation, which is a sign of acute colonic cell damage (Figure 2). Previous studies on the impact of probiotics on lipid peroxidation found that probiotics lowered MDA levels, which is consistent with the findings of this study [37,38]. The findings of our study showed that *P. freudenreichii* treatment alleviated the oxidative stress caused by toxic AOM metabolites.

The anti-cancer effects of *P. freudenreichii* are still being studied, and the exact mechanisms are not fully understood. However, several potential mechanisms have been proposed, including the detoxification of carcinogens, production of anti-oxidant compounds such as exopolysaccharides, induction of apoptosis in cancer cells, inhibition of tumor cell growth and migration, modulation of the immune system, an increase in the diversity and richness of the gut microbiota, prevention of the growth and proliferation of harmful bacteria, and production of SCFAs, such as propionate and butyrate, which can have anti-cancer effects. For example, butyrate has been shown to reduce inflammation in the gut and potentially prevent the development of cancer [19,28,29].

The use of probiotics is one of the strategies that influences the microbial gut diversity. Gut microbiota is being studied as a potential new approach to prevent or treat colorectal cancer. The consumption of certain probiotics helps to maintain the intestinal microbial balance by colonizing the human colon and preventing non-beneficial bacteria from colonizing [39].

In this study, the inoculation of AOM-induced rats with *P. freudenreichii* resulted in an enrichment of *Lactobacillus, Bifidobacterium*, and *Lactobacillus* LigA, all of which are beneficial microorganisms that help to maintain a healthy gut and intestinal flora balance. Although there was no significant difference in genus diversity between the groups, the *P. freudenreichii* group had a more diverse microbiome than the AOM group, implying that AOM inhibited gut microbiome diversification and resulted in a lower population of beneficial microorganisms in rats (Figure 6 and Figure 7). The findings of this study are in line with those of other studies that assessed the effect of probiotic consumption on gut microbial diversity [40,41,42]. Therefore, regular consumption of Swiss cheese and probiotic formulations would be beneficial, especially in people with a higher risk of colon polyps or suffering from CRC.

It is important to note that more research is needed to fully understand the effects of *P. freudenreichii* on different cell types and to determine the optimal strains and dosages for potential therapeutic applications. The limitations of this study include the lack of gene expression analysis of CRC markers, and limited molecular analysis of signaling pathways. Further studies are required to examine the underlying molecular mechanisms of *P. freudenreichii* and the effects of its metabolites on the expression of cancer marker genes to fully elucidate the preventive role of this beneficial microorganism.

## 4. Materials and Methods

### 4.1. Growth Conditions and Bacterial Strains

*Propionibacterium freudenreichii* (DSM 2027) was obtained from German collection of Microorganisms and Cell Cultures GmbH, Leibniz Institute DSMZ, Germany. The bacteria were freeze-dried and packaged in a double glass vial. To revive the freeze-dried bacterium cells, the bacteria were grown at 37 °C for 3 days in an anaerobic chamber using anaerogen gas packs (Thermo Fisher Scientific, Waltham, MA, USA) using reinforced clostridial media (RCM) broth (CM0149 Oxoid, Basingstoke, UK) and tryptic soy agar (TSA) enriched with sheep blood (Oxoid, Basingstoke, UK). The *P. freudenreichii* were washed twice with PBS and centrifuged at 5000× *g*. After that, bacteria were suspended at a concentration of 1 × 10^9^ CFU in PBS to create live bacterial inoculum. The cell-free supernatant was prepared by filtering the bacterial culture broth via a 0.22 µm polyethersulfone (PES) membrane and stored in single use aliquots at −20 °C until needed.

### 4.2. CRC Cell Line Culture and MTT Assay

HCT116 colorectal cancer cells were purchased from ATCC, USA. The HCT-116 cells were cultured in DMEM (Dulbecco’s modified Eagle’s medium) at 37 °C with 5% CO_2_ in a humidified oven. Antibiotics and 10% fetal bovine serum (FBS) were added to the media prior to subculture.

The antiproliferative effects of *P. freudenreichii* cell-free supernatant against HCT116 cells were tested using MTT cell viability assay. The cells were seeded overnight at 5000 cells per well before being exposed to the probiotic supernatant at serial concentrations (50%, 25%, 12.5%, 6.25%, and 3.13%). Then, 5-Fluorouracil was used as a standard chemotherapeutic drug control and prepared at various concentrations (50, 25, 12.5, 6.25, 3.13, 1.56, and 0.78 µg/mL), whereas cells treated with the vehicle where used as negative control. Plates were then incubated at 5% CO_2_ and 37 °C and for 72 h. After that, 10 μL MTT reagent were added to all wells before being re-incubated for another 4 h. The cells were lysed using 100 μL DMSO to expose the formazan crystal and the absorbance was measured at 570 nm [43]. The experiment was carried out in triplicate.

### 4.3. In Vivo Experiment

This experiment was conducted in accordance with institutional ethical guidelines, and the study was approved by the Institutional Animal Care and Use Committee with an IACUC reference ID: 2020-221205/PHARM/R/MAM (2019367). Twenty male SD rats (9–10 weeks) were maintained in IVC cages (3 and 2 rats per cage) under standard conditions with temperature of 22 °C, humidity 75%, and 12 h of light/dark cycle. The rats were fed on a standard rat pellet ad libitum. The rats were placed into four groups, each group with five rats, including a control group. The vehicle (phosphate buffered saline) was given to group 1, which is the normal control. Rats in groups 2–4 were injected subcutaneously with azoxymethane (Sigma-Aldrich in St. Louis, MO, USA) at a dose of 7 mg/kg body weight once a week for three weeks to induce aberrant crypt foci formation, and treated with *P. freudenreichii* (1 × 10^9^ CFU/mL) and 5-fluorouracil (35 mg/kg) for five weeks as described in (Table 2) [44].

### 4.4. Aberrant Crypt Foci (ACF) Examination

After five weeks of treatment, rats were sacrificed under anesthesia and colon was collected from all animals. The fixed colon tissues were stained with methylene blue 0.5% solution for 5 min and a macroscopic examination of the colon was done according to Bird’s method [45]. Aberrant crypt foci were examined and counted following a protocol described by McGinley et al. [46]. Briefly, the sum of ACF in the entire colon (from the distal to the proximal end) was counted in multiple fields (2-cm segments) for each sample and the data are presented as an average of 5 rats per group. The ACF were distinguished from the normal crypts by their larger size, existence of pericryptal zone, and their extended distance from basal to lamina surface of cells. The number and multiplicity of ACF on each colon segment were examined under light microscope.

### 4.5. Histological Examination of Colon

The harvested colon tissues from all animals were fixed in buffered formalin (10%) and histologically processed and embedded in paraffin wax. Eosin and hematoxylin were used to stain the colon sections. Under a light microscope, every ACF found in the colon of the treated groups was examined for its characteristics and compared to a normal untreated control [47].

### 4.6. DNA Extraction

To investigate the influence of probiotics on the diversity of rats’ gut microbiota, the total microbial DNA was extracted from fecal materials using DNA Stool QIAamp kit (Qiagen, Hilden, Germany) and the procedures were conducted based on to the manufacturer’s instructions [48]. The quality of DNA was monitored using 1% agarose gels and NanoDrop, then the concentration was adjusted to 1 µg/mL using nuclease-free water [49].

### 4.7. 16 S rRNA Sequencing Analysis

To characterize the gut microbiota of treated and untreated rat, 16 S rRNA sequencing was carried out for fecal specimens obtained from rats. The forward primer 515F (5′-GTGCCAGCMGCCGCGGTAA-3′) and reverse primer 806 R (5′-GGACTACHVGGGTWTCTAAT-3′) were used as a template to amplify the distinct region V4 of bacterial 16 S rRNA. A total of 20 μL of mixture was prepared containing 5 × FastPfu Buffer (4 μL), 5 μM of each primer (0.8 μL), 2.5 mM dNTPs (2 μL), FastPfu polymerase (0.4 μL), and 10 ng of template DNA. The PCR protocol of amplification started with initial denaturation for 2 min at 95 °C, followed by denaturation for 30 s at 95 °C each cycle for 25 cycles, annealing step for 30 s at 55 °C, extension step for 30 s at 72 °C, and a final extension which was carried out at 72 °C for 5 min. Using QIIME2 software package (version 1.9.1; http://qiime.org/scripts/assigntaxonomy.html, accessed on 19 June 2021), raw FASTQ 16S rRNA gene sequencing reads were demultiplexed and quality-filtered based on overlapping relationships; paired-reads were merged into a single read. Filtering was done using the following criteria: 300-bp reads were truncated at any site with an average quality score of 20 over a 50 bp sliding window, and reads shorter than 50 bp were discarded. Exact barcode matching, two nucleotide mismatches in primer matching, and reads with ambiguous characters were removed, and only sequences that overlapped for more than 10 bp were assembled based on their overlapping sequences. The reads were clustered using Functional Gene Database (FGR; Release 7.3; http://fungene.cme.msu.edu/, accessed on 19 June 2021) and GreenGenes database (release 13.5; http://greengenes.secondgenome.com/, accessed on 19 June 2021). Reads that could not be arranged were excluded. The merged reads were used for operational taxonomic units (OTUs) clustering, taxonomy classification, and assessing community diversity using UPARSE (version 7.1; http://drive5.com/uparse/, accessed on 19 June 2021), and chimeric sequences were identified and removed using UCHIME. The microorganism community was used to compare the similarity or dissimilarity of different groups and the correlation between both microbial development and environmental factors, as well as phylogenetic analysis and alpha and beta analysis diversity analysis [50].

### 4.8. Measurement of Lipid Peroxidation and Oxidative Stress

After harvesting the organs, colon samples were rinsed with cold PBS. Colon homogenates (10% *w*/*v*) were homogenized in cold PBS (pH 7.4) using an ice homogenizer. The homogenate samples were centrifuged at 4500 rpm for 15 min at 4 °C in a chilled centrifuge and then cell debris was removed. After that, the protein contents were measured and the supernatants were utilized to estimate the level of lipid peroxidation using malondialdehyde kit (Cayman Chemical, MI, USA). Briefly, to the samples/standards, SDS solution was added. The mixture was then topped with 4 mL of color reagent and mixed properly. For 1 h, the standard and samples solution tubes were submerged in boiling water. To inhibit the reactions, all tubes were incubated in an ice bath for 10 min. After that, all samples were centrifuged at 1600× *g* for 10 min at 4 °C. A 96-well plate was loaded with duplicate samples or standards, and the absorbance was measured with a plate reader at 532 nm.

### 4.9. Statistical Analysis

Statistical Package for the Social Sciences, version 27.0 for Windows was utilized to conduct the statistical analyses. One-way analysis of variance (ANOVA), followed with Tukey’s multiple comparisons post hoc test, was used to determine the statistical significance of the findings. For the in vitro parameters, the data were expressed as mean ± standard error mean (SEM) of three biological replicates, while, for the in vivo experiment, the data were presented as mean ± SEM of five rats. Results were considered statistically significant with a *p*-value of less than 0.05.

## 5. Conclusions

This study opens new possibilities for expanding the utilization of *P. freudenreichii*. It is generally regarded as a harmless bacterial probiotic by the FDA and is currently utilized as one of the strains in some probiotic supplement formulations. It is used in the production of cheese and can also be employed in the production of other dairy products, such as milk and yoghurt. Our study highlights the potential preventive effects of *P. freudenreichii* against colorectal cancer, which may encourage further investigation of its bioactive metabolites using a metabolomics approach to understand the molecular mechanisms underlying its effects. Furthermore, additional clinical trials using *P. freudenreichii* should be conducted to validate its health benefits and assess its clinical efficacy in patients with colorectal cancer.

## Figures and Tables

**Figure 1 ijms-24-08080-f001:**
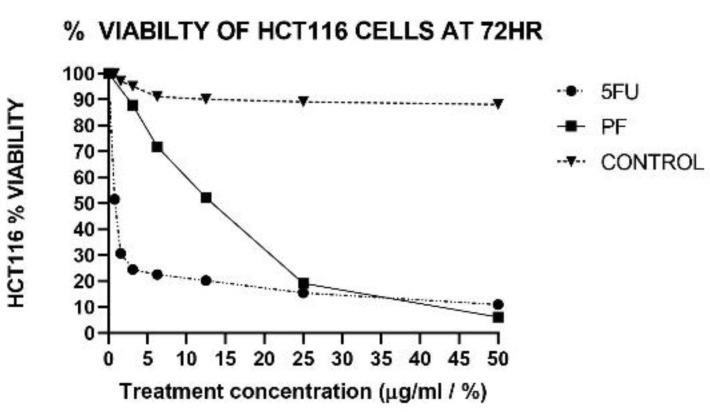
HCT116 cancer cell viability treated with cell-free supernatant of *P. freudenreichii* (PF) and 5-Fluorouracil (5-FU) for 72 h. Data are expressed as mean ± SEM of three replicates.

**Figure 2 ijms-24-08080-f002:**
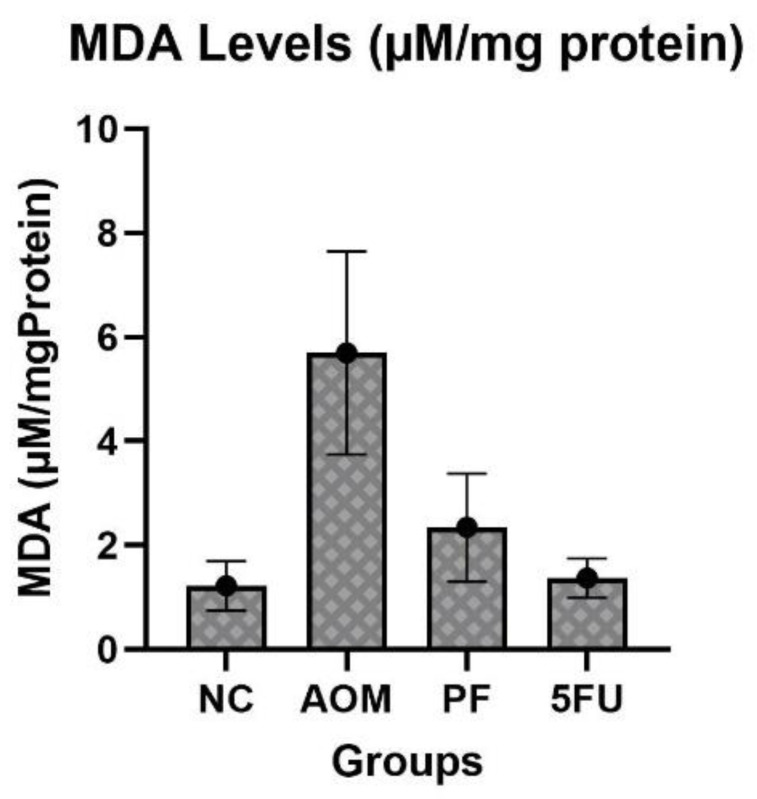
Oxidative stress levels in treated and untreated animals. Data were expressed as mean ± SEM. (AOM): azoxymethane, (PF): *P. freudenreichii*, (5-FU): 5-Fluorouracil, and (NC): normal control.

**Figure 3 ijms-24-08080-f003:**
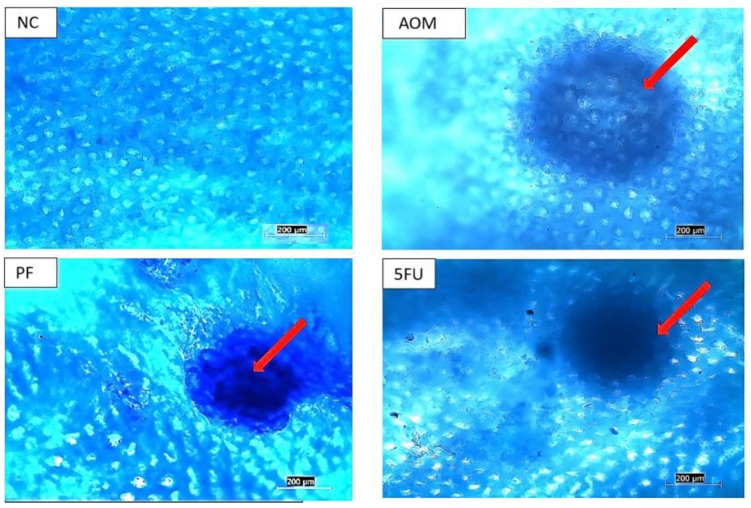
Aberrant crypt foci (ACF) in the colon of treated and control rats. (NC): normal control, (AOM): azoxymethane, (PF): *P. freudenreichii*, and (5-FU): 5-Fluorouracil, (scale bar = 200 µm). The red arrows point to the abnormal crypts (ACF lesions).

**Figure 4 ijms-24-08080-f004:**
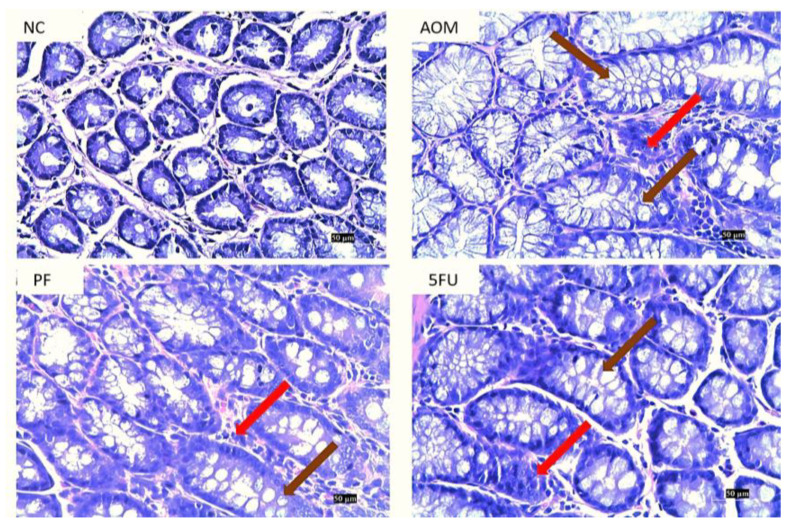
Histological analysis of the colons of treated and untreated rats. (NC): normal control, (PF): *P. freudenreichii*, (AOM): azoxymethane, and (5-FU): 5-Fluorouracil, (scale bar = 50 µm). The brown arrows indicate the enlarged crypts, and the red arrows show the infiltration of inflammatory cells.

**Figure 5 ijms-24-08080-f005:**
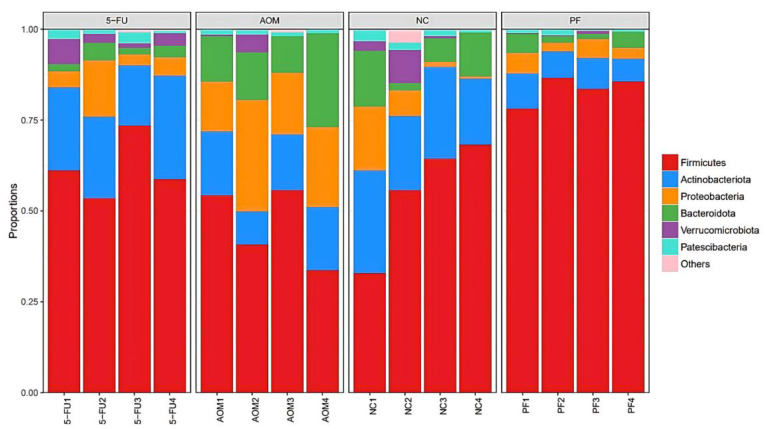
Plot of the gut microbiota diversity at Phylum level. (NC): normal control group, (PF): *P. freudenreichii* group, (AOM): azoxymethane control group, and (5FU): 5-fluorouracil reference group.

**Figure 6 ijms-24-08080-f006:**
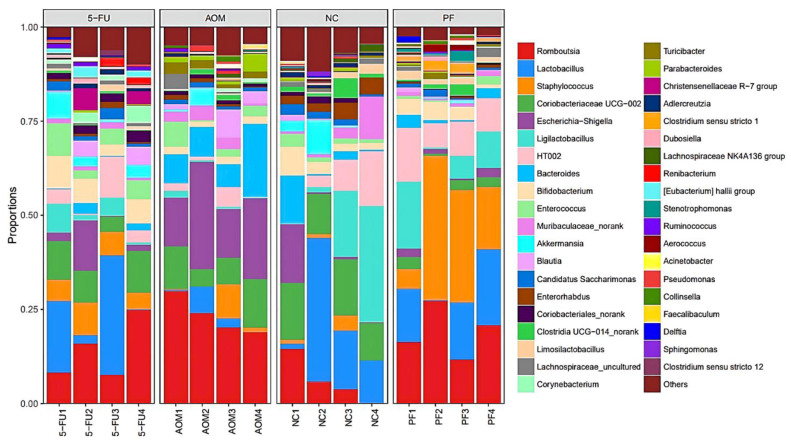
Plot of the gut microbiota diversity at genus level. (NC): normal control group, (PF): *P. freudenreichii* group, (AOM): azoxymethane control group, and (5FU): 5-fluorouracil reference group.

**Figure 7 ijms-24-08080-f007:**
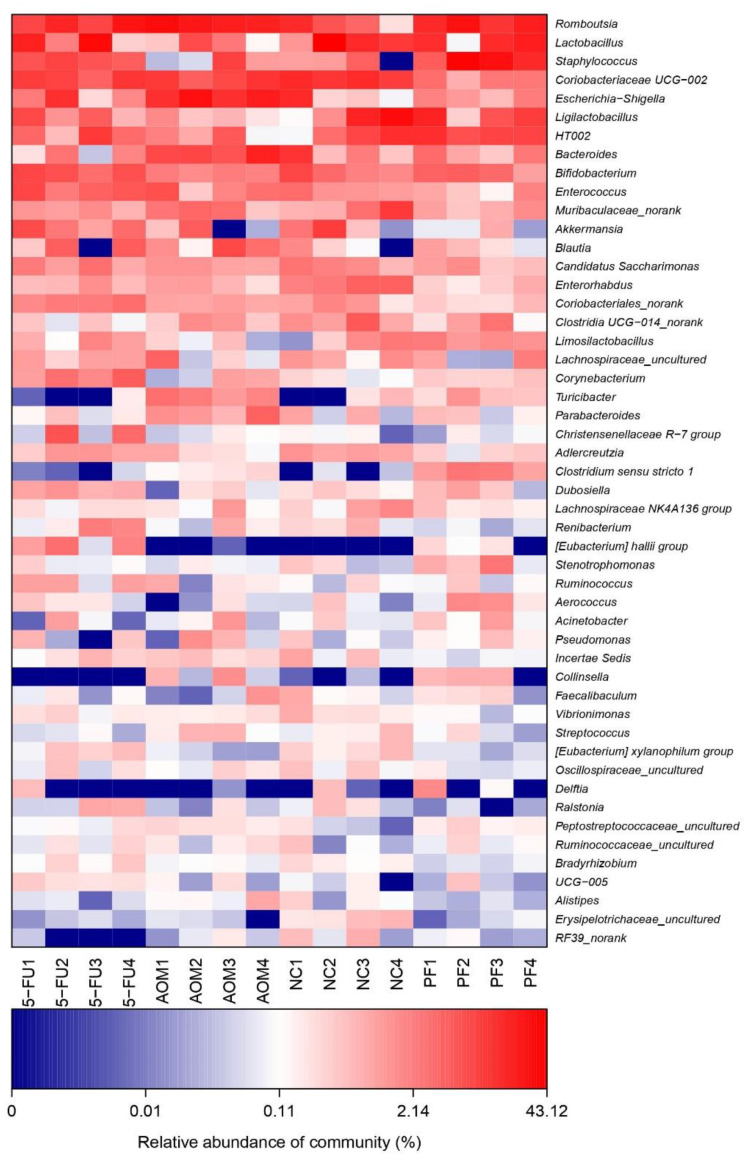
Heatmap analysis of microbial composition at genus level. (NC): normal control group, (PF): *P. freudenreichii* group, (AOM): azoxymethane control group, and (5FU): 5-fluorouracil reference group.

**Figure 8 ijms-24-08080-f008:**
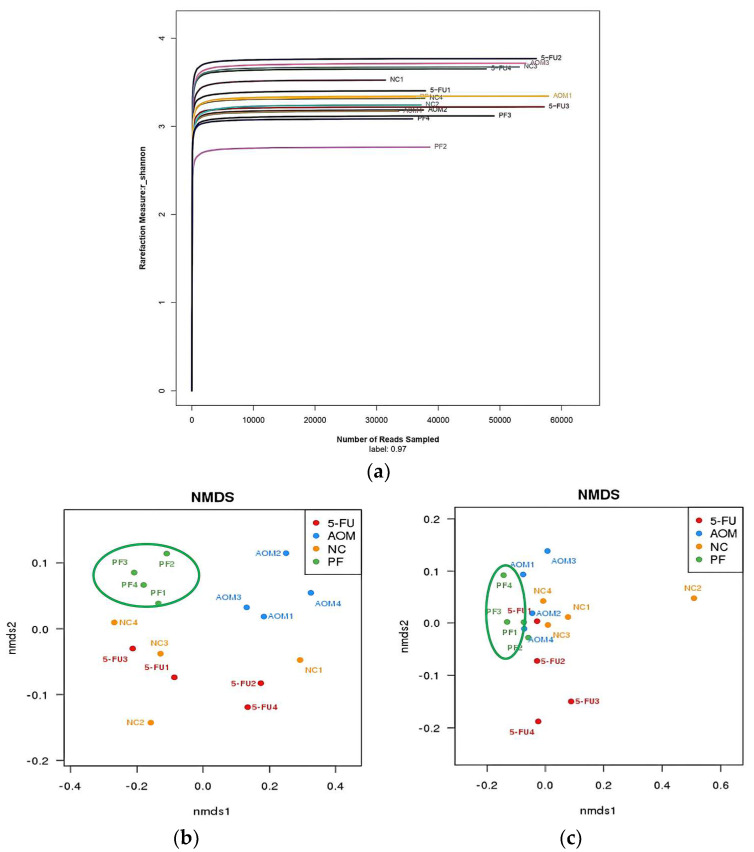
(**a**) Alpha diversity analysis using the Shannon index shown as sequences per sample. The x-axis shows the number of reads per sample, while the y-axis indicates the Shannon index. Each curve represents a different sample and is illustrated in a different color. (**b**,**c**): beta diversity analysis visualized with non-metric multidimensional scattering plot (NMDS), (NC): normal control group, (PF): *P. freudenreichii* group, (AOM): azoxymethane control group, and (5FU): 5-fluorouracil reference group.

**Table 1 ijms-24-08080-t001:** Aberrant crypt foci (ACF) count in treated and control groups.

Groups	1 Crypt	2 Crypts	3 Crypts	4 Crypts	5 Crypts & More	Total ACF	% Inhibition
Normal control	0	0	0	0	0	0	0
AOM group	9.8 ± 1.28	4.5 ± 0.77	8.0 ± 1.30	4.4 ± 0.40	7.2 ± 0.37	33.9 ± 3.18	0
*P. freudenreichii* group	5.2 ± 1.46	2.66 ± 0.68	2.25 ± 0.42	2.0 ± 0.31	2.8 ± 0.58	14.9 ± 2.73 ***	56.0 ***
5-FU group	2.75 ± 0.75	2.25 ± 0.75	1.75 ± 1.18	2.0 ± 0.40	2.25 ± 0.85	11.0 ± 1.10 ***	67.6 ***

Values were expressed as mean ± SEM, value with *** *p* < 0.001 indicates significance compared to azoxymethane (AOM) group.

**Table 2 ijms-24-08080-t002:** Grouping of experimental animals.

Groups	Treatment
Group 1 Normal control (NC)	Oral gavage of phosphate buffered saline (vehicle) 5 days/week for 5 weeks
Group 2 Tumor induced control (AOM)	7 mg/kg Azoxymethane (AOM), (s.c) once a week for 3 weeks + phosphate buffered saline (oral gavage, 5 days/week) starting one week prior AOM injection and continued for another 5 weeks
Group 3 *P. freudenreichii* (PF)	7 mg/kg AOM (s.c) once a week for 3 weeks + 1 × 10^9^ CFU/mL of *P. freudenreichii* (oral gavage, 5 days/week) starting one week prior AOM injection and continued for another 5 weeks
Group 4 5-Fluorouracil (5-FU)	7 mg/kg AOM (s.c) once a week for 3 weeks + 5-fluorouracil (35 mg/kg, IP, 3 times/week) for 5 weeks

## Data Availability

The data of 16S rRNA sequences were deposited into BioSample database, NCBI with accessions ID: SAMN25826982 to SAMN25826989, and SAMN25826998 to SAMN25827005. Web link: https://www.ncbi.nlm.nih.gov/bioproject/805078 (accessed on 30 December 2022).

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
