# Peer review of "Investigation into the Potential Role of Propionibacterium freudenreichii in Prevention of Colorectal Cancer and Its Effects on the Diversity of Gut Microbiota in Rats"

_ijms, 2023, doi:10.3390/ijms24098080_

Round 1
Reviewer 1 Report
The manuscript entitled “Investigation into the potential role of Propionibacterium freudenreichii in prevention of colorectal cancer and its effects on the diversity of gut microbiota in rats” is about to provide the function of P. freudenreichii as a probiotics in rats.
This manuscript was written well and experiments were done properly, but need some modification to be published.
At line 33, “In vitro study showed that HCT116 cells were suppressed by the probiotic cell free supernatant with an IC50 value of 13.3±0.133.” this should be the growth of HCT116 cells or viability.
It needs to modify because the meaning is complicated at the sentence of “In recent years, colorectal cancer (CRC) has accounted for 10% of all cancer diagnoses and cancer-related deaths globally, with a mortality rate around 25% lower than men.”
I strongly recommend the editing of this manuscript.
At line 87 89, and 91, P. freudenreichii must be italicized, and there are many in the manuscript.
Authors showed the alleviated level of oxidative stress by PT treatment. However, that is usually increased in cancer cell, authors need explanation of this relationship in discussion. Is that because of depletion of enzyme or changing the expression level?
Authors showed the Aberrant crypt foci at table 1 and figure 2. at table 1, authors provide the number of Aberrant crypt foci. but at the figure there is only one of aberrant crypt foci. If authors want to provide the number of aberrant crypt foci, authors need provide the gigger size figure that shown more aberrant crypt foci. and additionally, is there any other difference of aberrant crypt foci or other cells?
Authors showed anticancer activity by in vitro but not in vivo. In vivo experiment showed oxidative stress level, aberrant crypt foci formation and gut microbiota. Is there enough rationale that PF showed prevention of cancer or no? and moreover, the relation between P. freudenreichii with cancer is already known, I am not sure what is the novelty of this manuscript. Authors also did prevention experiments with mouse. I think authors rather talk about the prevention of cancer formation than the inhibition of the growth of cancer cells in the whole manuscript. For example, first experiments make readers think about the effects of PF as an anticancer agent even though title showed prevention.
Authors need to provide how the 16S rRNA analyzed after PCR.
Authors need double check the references.
Author Response
Dear Editor,
Thank you for your prompt handling of our recently submitted manuscript, with ID: ijms-2167398, titled "Investigation into the potential role of Propionibacterium freudenreichii in prevention of colorectal cancer and its effects on the diversity of gut microbiota in rats"
The response to the reviewers’ comments is as given below and the corrections are highlighted in the revised manuscript.
REVIWER 1:
- At line 33, “In vitro study showed that HCT116 cells were suppressed by the probiotic cell free supernatant with an IC50 value of 13.3±0.133.” this should be the growth of HCT116 cells or viability.
REPLY: The antiproliferative activity of P. freudenreichii cell-free supernatant against HCT116 cells was expressed using the IC50 value because this best describes how much of the probiotic supernatant is needed to inhibit the viability of HCT 116 colorectal cancer cells, thus providing a measure of potency. In addition, a graph of the % viability of HCT 116 cells was also provided on page 3. In the Abstract, the sentence has been corrected accordingly (page 1).
- It needs to modify because the meaning is complicated at the sentence of “In recent years, colorectal cancer (CRC) has accounted for 10% of all cancer diagnoses and cancer-related deaths globally, with a mortality rate around 25% lower than men.” I strongly recommend the editing of this manuscript.
REPLY: The paragraph has been amended (page 1). The language of the manuscript has been edited by a native speaker.
- At line 87, 89, and 91, P. freudenreichii must be italicized, and there are many in the manuscript.
REPLY: The paper has been edited and P. freudenreichii has been properly italicized throughout the manuscript.
- Authors showed the alleviated level of oxidative stress by PF treatment. However, that is usually increased in cancer cell, authors need explanation of this relationship in discussion. Is that because of depletion of enzyme or changing the expression level?
REPLY: Our experimental model is a prevention study in which we induced precancerous lesions (colonic aberrant crypt foci (ACF)) using a carcinogenic compound, azoxymethane (AOM). AOM is metabolized by cytochrome into a toxic metabolite (methylazoxymethanol) that causes oxidative stress, DNA mutations, and neoplastic lesions. Oxidative stress in normal colon cells is mediated by glutathione depletion, which impairs the total antioxidant capacity in colonic cells. The findings of our study showed that P. freudenreichii treatment alleviated the oxidative stress caused by toxic AOM metabolites. This has been highlighted and explained further on page 10.
- Authors showed the Aberrant crypt foci at table 1 and figure 2. at table 1, authors provide the number of Aberrant crypt foci. but at the figure there is only one of aberrant crypt foci. If authors want to provide the number of aberrant crypt foci, authors need provide the bigger size figure that shown more aberrant crypt foci. and additionally, is there any other difference of aberrant crypt foci or other cells?.
REPLY: Table 1 represents the sum of ACF in the entire colon (from the distal to the proximal end), and the count was conducted in 2-cm segments using a lower power lens by counting multiple fields for each sample and the data are presented as an average of 5 rats per group. In Figure 3, we used a higher-power lens to show the number of crypts and detail per foci.
There are differences between ACF and normal cells because aberrant crypt foci are commonly recognized as early indicators of human large intestine neoplasms and ACF development is now commonly acknowledged as a viable endpoint in short-term chemopreventive studies and is considered the gold standard of colon carcinogenesis biomarkers. ACFs are distinguished from normal crypts by their larger size, existence of pericryptal zone, and extended distance from basal to lamina surface of cells. This has been highlighted in the methodology section (page 12).
- Authors showed anticancer activity by in vitro but not in vivo. In vivo experiment showed oxidative stress level, aberrant crypt foci formation and gut microbiota. Is there enough rationale that PF showed prevention of cancer or no? and moreover, the relation between P. freudenreichii with cancer is already known, I am not sure what is the novelty of this manuscript. Authors also did prevention experiments with mouse. I think authors rather talk about the prevention of cancer formation than the inhibition of the growth of cancer cells in the whole manuscript. For example, first experiments make readers think about the effects of PF as an anticancer agent even though title showed prevention.
REPLY: Thank you for the comment, in our experiment we are showing the preventive effects of P. freudenreichii against the carcinogenic effects of AOM by inhibiting the ACF initiation which is an early indicator of colon neoplasms. On the other hand, the in vitro study showed that the supernatant of PF exerts anti-proliferative properties against HCT 116 cells.
The novelty is that this is the first study on the anti-proliferative effects of PF against HCT116 colorectal cancer cells. Another in vivo study was carried out in rats, but they used another carcinogen, which was only administered once. However, our study improves on this by challenging rats with repeated doses of AOM, which is more toxic. Our results confirmed the findings of a previous study on the inhibition of ACF formation and also showed that PF modulates the gut microbiota by enhancing its richness and abundance of gut microbiota.
We have amended some paragraphs to replace the “anti-cancer activity” with “prevention” and the final conclusion stated that “The findings of this study raveled the potential preventive effects of this probiotic against colorectal cancer in rats and its anti-proliferative effects CRC cell line” (page 14), and this was also reflected in the title of the manuscript.
- Authors need to provide how the 16S rRNA analyzed after PCR.
REPLY: The 16S rRNA analysis has been added on page 13.
- Authors need double check the references.
REPLY: The references have been checked and corrected accordingly.

Reviewer 2 Report
Overall the article titled "Investigation into the potential role of Propionibacterium freudenreichii in the prevention of colorectal cancer and its effects 3 on the diversity of gut microbiota in rats" lacks novelty and a preliminary indication that Propionibacterium freudenreichii may have potential to prevent colorectal cancer. Similar reports are already available.
1) what is the novelty of the article
2) Is the P. freudenreichii a probiotic strains? any reference from the previous studies?
3) No detailed analysis of the alpha and beta diversity of microbiota was provided
4) Bioinformatic analysis done is missing
5) Molecular mechanisms and gene expression of cancer markers are also missing, which can strengthen the article.
6) Further comments are available in PDF file

Author Response
Dear Editor,
Thank you for your prompt handling of our recently submitted manuscript, with ID: ijms-2167398, titled "Investigation into the potential role of Propionibacterium freudenreichii in prevention of colorectal cancer and its effects on the diversity of gut microbiota in rats"
The response to the reviewers’ comments is as given below and the corrections are highlighted in the revised manuscript.
REVIWER 2
- What is the novelty of the article?
REPLY: The novelty of this study is that using a rat model, we showed that the probiotic P. freudenreichii can prevent the initiation of CRC by inhibiting ACF formation and modulating the gut microbiota by improving the diversity and richness of beneficial microbiota. This is novel because other studies have mostly only studied this probiotic using cancer cell lines other than HCT116 colorectal cancer cells and even animal studies that induced cancer have used only a single dose. This work shows that even with repetitive AOM challenge in the rat colon, PF was still able to reduce the multiplicity and rate at which ACF formed thus establishing its preventative properties.
- Is the P. freudenreichii a probiotic strain? any reference from the previous studies?
REPLY: P. freudenreichii is used as a cheese-ripening starter and as a probiotic that is well-known for its long history of consumption in Swiss type cheeses. It has been established as a probiotic strain because it falls under the category of a “Living organism that, when established in adequate amounts, confers health benefits to the host,” as described by the FDA. References 9 - 14 show this.
- Thierry, A., S.-M. Deutsch, H. Falentin, M. Dalmasso, F. J. Cousin and G. Jan. "New insights into physiology and metabolism of propionibacterium freudenreichii." International journal of food microbiology 149 (2011): 19-27.
- Cousin, F. J., B. Foligné, S.-M. Deutsch, S. Massart, S. Parayre, Y. Le Loir, G. Boudry and G. Jan. "Assessment of the probiotic potential of a dairy product fermented by propionibacterium freudenreichii in piglets." Journal of agricultural and food chemistry 60 (2012): 7917-27.
- Lan, A., A. Bruneau, C. Philippe, V. Rochet, A. Rouault, C. Hervé, N. Roland, S. Rabot and G. Jan. "Survival and metabolic activity of selected strains of propionibacterium freudenreichii in the gastrointestinal tract of human microbiota-associated rats." British Journal of Nutrition 97 (2007): 714-24.
- Jan, G. "Belzacq as, haouzi d, rouault a, metivier d, kroemer g, and brenner c." Propionibacteria induce apoptosis of colorectal carcinoma cells via short-chain fatty acids acting on mitochondria. Cell Death Differ 9 (2002): 179-88.
- Lan, A., D. Lagadic-Gossmann, C. Lemaire, C. Brenner and G. Jan. "Acidic extracellular ph shifts colorectal cancer cell death from apoptosis to necrosis upon exposure to propionate and acetate, major end-products of the human probiotic propionibacteria." Apoptosis 12 (2007): 573-91.
- Lan, A., A. Bruneau, M. Bensaada, C. Philippe, P. Bellaud, S. Rabot and G. Jan. "Increased induction of apoptosis by propionibacterium freudenreichii tl133 in colonic mucosal crypts of human microbiota-associated rats treated with 1, 2-dimethylhydrazine." British Journal of Nutrition 100 (2008): 1251-59.
- No detailed analysis of the alpha and beta diversity of microbiota was provided
REPLY: The methodology of alpha and beta diversity analyses have been added on page 13. In addition, Figure 8 has been added (page 9) to visualize beta diversity using a non-metric dimensional scattering plot (NMDS).
- Bioinformatic analysis done is missing:
REPLY: We have included details of the bioinformatics analysis on page 13.
- Molecular mechanisms and gene expression of cancer markers are also missing, which can strengthen the article.
REPLY: We acknowledge that molecular mechanisms and gene expression studies could strengthen our study. Therefore, we have added that as one of the limitations of this study, further investigations by our team will address this in the future (page 11).
- Further comments are available in PDF file
REPLY: Thank you, all the corrections have been applied accordingly.

Round 2
Reviewer 1 Report
Authors did not check the references.
Lots of reference do not have the name of journal, and some have abbreviated name but others full name.
Author Response
Thank you for your comment. We apologize for the issue with the citation manager. The references have been thoroughly reviewed, and the journal names have now been included.
Reviewer 2 Report
Still lack scientific soundness
Author Response
Thank you for your constructive feedback. We appreciate it and take your comments seriously. Regarding your concern about the lack of scientific soundness, we would like to clarify that our study was conducted in accordance with established experimental protocols and guidelines, and all data were rigorously analyzed and interpreted. We understand that there may be areas for improvement, and we have made revisions to the manuscript to ensure that our results, discussion and conclusions are robust and well-supported by the data.
We have also improved our introduction by providing more information on the potential role of probiotics and their bioactive metabolites, such as short chain fatty acids (SCFA), propionate, and acetate, in preventing colorectal cancer, which is the focus of this study. Probiotics can also impact the host microbiome, influencing the relationship between diet, inflammation, and cancer. Our study highlights the potential role of P. freudenreichii and may inspire other scientists to investigate its bioactive metabolites using a metabolomics approach to understand the molecular mechanisms against colorectal cancer. The amendments in the introduction and conclusion have been highlighted (page 2 and 14).
Round 3
Reviewer 2 Report
Manuscript has improved extensively but authors should still add
1) Claim that the P. freudenreichii strain used in this study is probiotics. Unless it is characterized and its probiotic prerequisits are established it can notbe considered as probiotics. Since probiotic properties are strain specific and not the specie specific, therefore the characterization of this particular strain or a previous study on it will augment the claims.
2) What about the safety profile of this particular strain in terms of transferable antibiotic resistance
3) what about the cytotoxicity of the strain for other normal intestinal epithelial cells, if the strain is cyctotoxic for cancer cells what about its effects on the normal cells.
4) Elaborate the possible mechanisms of anti-cancer effect of this particular strain
5) results still need to be tone down and limitations of the study should also be given in discussion part.
Author Response
17th February 2023
Dear Editor,
I am writing to express my gratitude for your prompt handling of our recently submitted manuscript with the ID: ijms-2167398, titled "Investigation into the potential role of Propionibacterium freudenreichii in the prevention of colorectal cancer and its effects on the diversity of gut microbiota in rats."
We appreciate the constructive feedback provided by the reviewer, and we have taken their comments into consideration when revising our manuscript. We have provided a detailed, point-by-point response to each comment, and we believe that the changes we have made adequately address the concerns raised by the reviewer. The response to the reviewer’s comments is given below, and the revisions have been highlighted in the revised manuscript.
REVIWER 2
- Claim that the P. freudenreichii strain used in this study is probiotics. Unless it is characterized and its probiotic prerequisits are established it cannot be considered as probiotics. Since probiotic properties are strain specific and not the specie specific, therefore the characterization of this particular strain or a previous study on it will augment the claims?
REPLY: Propionibacterium freudenreichii is widely recognized as a probiotic in the dairy industry and has been used for centuries as a ripening agent for cheese. Recent research has focused on exploring its potential as a probiotic, and we have included information on this topic in revised manuscript on page 2, specifically in lines 66-85.
- What about the safety profile of this particular strain in terms of transferable antibiotic resistance?
REPLY: Propionibacterium freudenreichii has been designated as “Generally Recognized as Safe” (GRAS) by the US Food and Drug Administration (USFDA). Additionally, P. freudenreichii has an intrinsic resistance to several antibiotics, including aminoglycosides, sulfonamides, 1st & 2nd-generation quinolones, oxacillin, and metronidazole. It is worth noting that this resistance is not conferred by plasmids. We included more information on page 2, lines 86-92.
- What about the cytotoxicity of the strain for other normal intestinal epithelial cells, if the strain is cytotoxic for cancer cells what about its effects on the normal cells.
REPLY: Research studies have demonstrated that P. freudenreichii does not have an adverse effects on normal intestinal epithelial cells. In fact, some studies have suggested that P. freudenreichii may have a protective effect on the intestinal epithelium by promoting cell differentiation and preventing inflammation, which can be beneficial for maintaining a healthy gut. It is important to note that more research is needed to fully understand the effects of P. freudenreichii on different cell types and to determine the optimal strains and dosages for potential therapeutic applications. We have added more details on this topic to page 10, lines 246-252.
- Elaborate the possible mechanisms of anti-cancer effect of this particular strain
REPLY: Thank you for your comment. We have provided information on the potential anti-cancer and preventative properties of P. freudenreichii on page 2, lines 95-99, and on page 10, lines 238-2245 and page 11, lines 292-300.
The anti-cancer effects of P. freudenreichii are still being studied, and the exact mechanisms are not fully understood. However, several potential mechanisms have been proposed, including, detoxification of carcinogens, production of anti-oxidant compounds such as exopolysaccharides, induction of apoptosis in cancer cells, inhibition of tumor cell growth and migration, modulation of the immune system, increase the diversity and richness of the gut microbiota, prevention of the growth and proliferation of harmful bacteria, and production of short-chain fatty acids (SCFAs), such as propionate and butyrate, which can have anti-cancer effects. For example, butyrate has been shown to reduce inflammation in the gut and potentially prevent the development of cancer.
- Results still need to be tone down and limitations of the study should also be given in discussion part.
REPLY: Thank you for your comment. We agree that the language used in discussing the results could be further toned down to better reflect the findings of our study. We revised the manuscript accordingly to make the language more appropriate. In addition, we elaborated more on the limitations to our study, including the lack of gene expression analysis of CRC markers, and limited molecular analysis of signaling pathways. These limitations will be addressed in our future studies and added in the revised manuscript page 11, lines 317-323.

Round 4
Reviewer 2 Report
Manuscript has been improved